# Effect of interventions incorporating personalised cancer risk information on intentions and behaviour: a systematic review and meta-analysis of randomised controlled trials

Juliet A Usher-Smith,[1] Barbora Silarova,[2] Stephen J Sharp,[2] Katie Mills,[1] Simon J Griffin[1]

[1]The Primary Care Unit, Department of Public Health and Primary Care, School of Clinical Medicine, University of Cambridge, Cambridge, UK
[2]MRC Epidemiology Unit, Institute of Metabolic Science, Cambridge, UK

**Correspondence to**
Dr Juliet A Usher-Smith;
jau20@medschl.cam.ac.uk

## ABSTRACT

**Objective** To provide a comprehensive review of the impact on intention to change health-related behaviours and health-related behaviours themselves, including screening uptake, of interventions incorporating information about cancer risk targeted at the general adult population.

**Design** A systematic review and random-effects meta-analysis.

**Data sources** An electronic search of MEDLINE, EMBASE, CINAHL and PsycINFO from 1 January 2000 to 1 July 2017.

**Inclusion criteria** Randomised controlled trials of interventions including provision of a personal estimate of future cancer risk based on two or more non-genetic variables to adults recruited from the general population that include at least one behavioural outcome.

**Results** We included 19 studies reporting 12 outcomes. There was significant heterogeneity in interventions and outcomes between studies. There is evidence that interventions incorporating personalised cancer risk information do not affect intention to attend or attendance at screening (relative risk 1.00 (0.97–1.03)). There is limited evidence that they increase smoking abstinence, sun protection, adult skin self-examination and breast examination, and decrease intention to tan. However, they do not increase smoking cessation, parental child skin examination or intention to protect skin. No studies assessed changes in diet, alcohol consumption or physical activity.

**Conclusions** Interventions incorporating personalised cancer risk information do not affect uptake of screening, but there is limited evidence of effect on some health-related behaviours. Further research, ideally including objective measures of behaviour, is needed before cancer risk information is incorporated into routine practice for health promotion in the general population.

## INTRODUCTION

In 2006, the US National Cancer Institute recognised risk-prediction models as an 'area of extraordinary opportunity'.[1] Since then, an increasing number of risk-prediction models have been developed. Such models can facilitate a personalised approach to cancer prevention and treatment and a more equitable and cost-effective distribution of finite resources by targeting screening and prevention activities at those most likely to benefit. Furthermore, being able to estimate, communicate and monitor individual risk and demonstrate the impact of lifestyle change on future risk of cancer may complement wider collective approaches to shifting population distributions of behaviour, risk factors and cancer risk.

Research has shown that many individuals have incorrect perceptions of their risk of cancer[2–4] and that both overestimation and underestimation are associated with maladaptive health-related behaviours.[5] Additionally, while up to 40% of all cancers are attributable

**Strengths and limitations of this study**

► This systematic review is the first comprehensive review of the effect on intention and health-related behaviour of individuals in the general population of interventions delivered across multiple settings which incorporate personalised information about cancer risk.
► The use of a broad search strategy across multiple databases enabled us to identify 19 randomised controlled trials reporting the impact of interventions incorporating personalised cancer risk information on 12 outcomes.
► However, there was large heterogeneity across the studies, including the content of interventions and the outcome measures. This meant that it was only possible to meta-analyse one outcome, attendance at screening and, in many studies, separating the effect of the risk information alone from additional elements of the interventions was not possible.

**BMJ**

to lifestyle factors,[6] only 3% of people are aware that being overweight can increase their risk of cancer and less than a third that physical activity could help reduce risk.[7–10] One in seven people additionally believes that lifetime risk of cancer is unmodifiable.[11] Most behaviour change theories suggest that perceived risk is important alongside other constructs such as self-efficacy and response efficacy in promoting behaviour change.[12 13] Providing individuals with estimates of their risk of cancer alongside other behaviour change interventions may therefore help motivate behaviour change at an individual level. It may also enable individuals to make more informed decisions about uptake of screening tests for cancer. This has led to the development of an increasing number of interventions incorporating information about cancer risk being developed.

Understanding the impact of interventions incorporating information about cancer risk on behaviour and intention to change behaviour before they are introduced into routine practice is important. Previous systematic reviews in this area have focused only on trials in primary care[14] or tailored information about cancer risk and screening.[15 16] In this review, we aimed to provide a comprehensive synthesis of the impact of interventions incorporating personalised information about cancer risk on intention to change health-related behaviours and health-related behaviours within the general adult population.

## METHODS
We performed a systematic literature review following an a priori established study protocol (available on request). Reporting followed the Preferred Reporting Items for Systematic Reviews and Meta-Analyses statement.[17]

### Search strategy
We performed an electronic literature search of MEDLINE, EMBASE, CINAHL and PsycINFO from January 2000 to July 2017 with no language limits using a combination of subject headings and free text incorporating 'cancer', 'risk/risk factor/risk assessment' and 'prediction/model/score/tool' (see online supplementary file 1 for the complete search strategies). We then extended the search by manually screening the reference lists of all included papers. We chose to begin the search in 2000 as the previous review of tailored information about cancer risk and screening had noted that computer-delivered interventions, as would be required for calculating risk scores, were only described in publications from 2000 onwards.[15]

### Study selection
We included studies if they were randomised controlled trials (RCTs) published as a primary research paper in a peer-reviewed journal, included adults with no previous history of cancer, included provision to individuals of a personal estimate of future cancer risk based on two or

more non-genetic variables, and reported at least one behavioural outcome. In order to focus on the provision of personalised cancer risk to the general population, we excluded studies which had recruited participants on the basis of a personal or family history of cancer or following referral to specialist cancer risk services. Vignette, before-and-after studies without a control group, cross-sectional, longitudinal and qualitative studies were also excluded along with conference abstracts, editorials, commentaries and letters.

Two reviewers (JAU-S and BS) each screened half of the titles and abstracts to exclude papers that were clearly not relevant. A third reviewer (SJG) independently assessed a random selection of 5% of the papers screened by each of the first reviewers. The full text was examined if a definite decision to exclude could not be made based on title and abstract alone. Two reviewers (JAU-S and BS) independently assessed all full-text papers. We discussed papers for which it was unclear whether or not the inclusion criteria were met at consensus meetings with a third reviewer (SJG). Papers written in languages other than English were translated into English for assessment and subsequent data extraction.

### Data extraction
Two researchers (JAU-S+BS/KM) independently extracted data from studies included in the review using a standardised data abstraction form to reduce bias. The data extracted included: (1) study characteristics (cancer type, study design, study setting or duration of follow-up), (2) selection of participants (inclusion criteria or method of recruitment/randomisation), (3) participant characteristics (age, level of cancer risk or sample size), (4) intervention (risk tool used, method and format of risk communication, additional information or follow-up provided) and (5) measured outcome(s). Reviewers were not blinded to publication details.

### Quality assessment
We conducted quality assessment at the same time as data extraction using a checklist based on the Critical Appraisal Skills Programme guidelines[18] as an initial framework. This includes eight questions concerning whether the study addressed a clearly focused issue, the method of recruitment and randomisation, whether blinding was used, the measurement of the exposure and outcome, the comparability of the study groups and the follow-up. Each study was then classified as high, medium or low quality. No studies were excluded based on quality alone.

### Data synthesis and statistical analysis
For analysis, we grouped the measured outcomes into those relating to: (1) preferences or intention to attend cancer screening, (2) cancer screening uptake, (3) intention or motivation to change health-related behaviour and (4) change in health-related behaviour. It was only possible to pool results for screening attendance. For this,

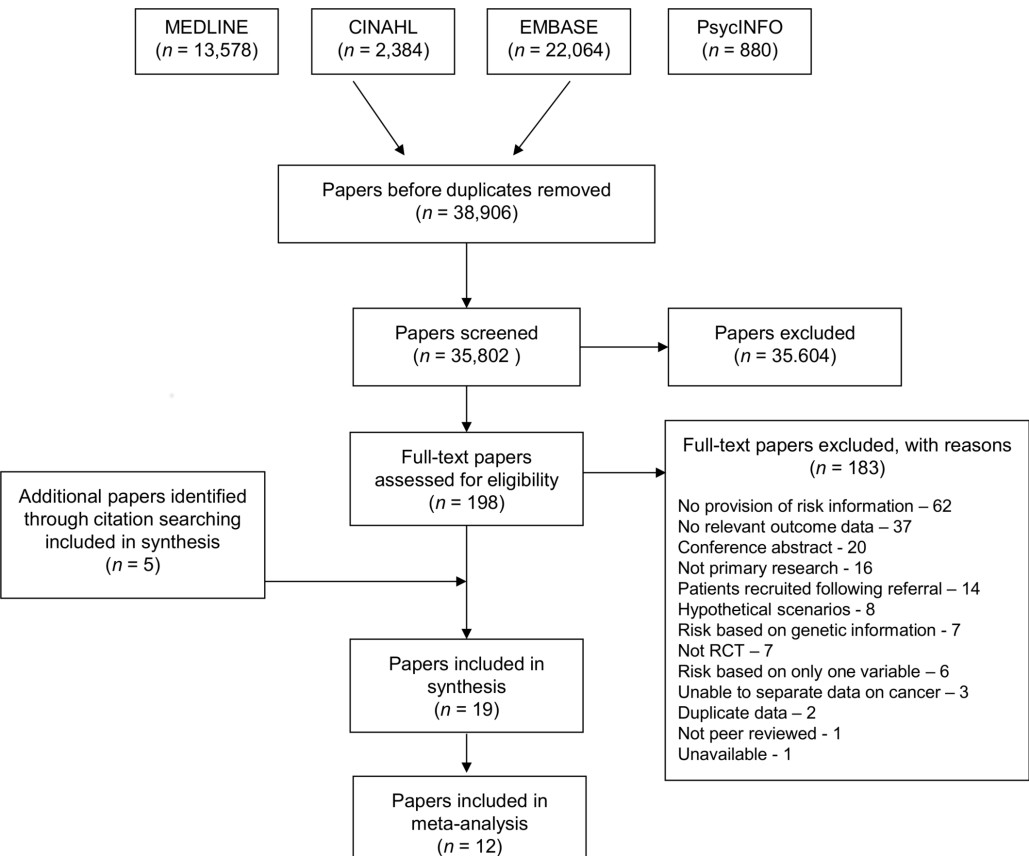

**Figure 1** Preferred Reporting Items for Systematic Reviews and Meta-Analyses flow diagram. RCT, randomised controlled trial.

we used random-effects meta-analysis[19] and the 'metan' package in STATA. We present intervention effects as relative risk (RR) rather than OR to avoid overestimating the risk.[20] We estimated the heterogeneity between studies using the $I^2$ statistic. All analyses were conducted using statistical software package STATA/SE V.12.

## RESULTS

After duplicates were removed, the search identified 38 906 papers. Of these, 35 604 were excluded at title and abstract level and a further 183 after full-text assessment. After title and abstract screening by the first reviewers (JAU-S and BS), no additional papers met the inclusion criteria in the random 5% screened by the second reviewer (SJG). The most common reasons for exclusion at full-text level were that the papers did not include provision of a personal risk estimate (n=62), did not include any data on predefined outcomes (n=37), were conference abstracts (n=20) or were not primary research (n=16) (figure 1). Five further papers were identified through citation searching, giving 19 studies included in the analysis.

A summary of the participants and setting of those 19 studies is shown in table 1. With the exception of three studies conducted in the UK,[21–23] all studies took place in the USA. Most recruited participants from those

attending primary care clinics (n=3) or lists of potentially eligible individuals from electronic medical records (n=7), telephone services (n=1), insurance records (n=1) or survey companies (n=1). Two recruited through schools, community centres and universities, one from those calling a cancer information service and three used public advertisements.

In eight studies personalised information was provided about risk of breast cancer, in five about risk of colorectal cancer (CRC), in three risk of skin cancer, one lung cancer, one cervical cancer and one multiple cancers. Further details of the risk models used to calculate the risk estimate provided to participants and the format of the intervention(s) are given in table 2. All eight studies providing personalised information about breast cancer risk used the Gail risk model.[24] This was the first risk model developed for breast cancer and includes age, age at menarche, age at first live birth, number of previous biopsies, number of biopsies showing atypical hyperplasia and number of first-degree relatives with breast cancer. Where details were given (n=3), all studies on CRC used the Harvard Cancer Risk tool[25] which includes family history, height and weight, alcohol consumption, vegetable and red meat consumption, physical activity, screening history, a history of inflammatory bowel disease and use of aspirin, folate and female hormones.

**Table 1** Details of the setting and key outcomes of the included studies

| Author, year | Cancer site(s) | Follow-up | Setting and participants | Risk level/comorbidities | Outcome(s) | Quality |
|---|---|---|---|---|---|---|
| Bodurtha et al,[31] 2009 | Breast | 18 months | 899 women with no history of breast cancer recruited from waiting rooms of four women's health clinics | Not given | Mammography, clinical breast examinations, breast self-examination, mammography intentions | M–H |
| Bowen et al,[41] 2006 | Breast | 6 and 24 months | 150 sexual minority women recruited via public advertisements | Mean Gail lifetime risk 12% | Breast self-examination, breast cancer screening | H |
| Bowen and Powers,[42] 2010 | Breast | 12 months | 1366 women recruited via purchased lists of telephone numbers with no previous diagnosis of breast cancer | Mean Gail lifetime risk 12% | Breast self-examination, mammography | M |
| Davis et al,[34] 2004 | Breast | 1 month | 392 women with no history of cancer calling the Cancer Information Service | 27% 2%–6% lifetime risk; 32% 6%–9% lifetime risk; 41% 9%–46% lifetime risk | Adherence to breast cancer screening, intention for breast cancer screening | M |
| Glanz et al,[45] 2013 | Skin | 16 weeks | Convenience sample of 1047 parents not currently being treated for skin cancer recruited through schools and community centres | 38% high risk | Sun protection habits, sun exposure, skin examination by parents | M |
| Glazebrook et al,[22] 2006 | Skin | 6 months | 589 recruited from 10 primary care practices from a convenience sample of appointments | Not given | Sun protection habits | M |
| Greene and Brinn,[44] 2003 | Skin | 3–4 weeks | 141 undergraduates at one university who received extra credit for participation | Not given | Intention to tan, actual tan bed usage | L–M |
| Helmes et al,[35] 2006 | Breast | 3 months | Random sample of 340 members of state healthcare system with no history of breast/ovarian cancer or testing for cancer risk | Mean 9.5% (3.2) lifetime risk | Intention to have mammogram and clinical breast examination, intention to do breast self-examination | M |
| Holloway et al,[21] 2003 | Cervical | 0 and 4 years | 1890 women attending routine cervical smear test at one of 29 general practitioner practices | 78%–80% very low risk; 20%–22% low risk | Preference for future screening interval, actual screening behaviour | M–H |
| Lipkus and Klein,[30] 2006 | Colorectal | 0 | 160 members of general public with no history of CRC or screening for CRC recruited through newspaper advertisements | Not given | Ambivalence, intention to screen using an FOBT, actual FOBT screening rates | M |

Continued

**Table 1** Continued

| Author, year | Cancer site(s) | Follow-up | Setting and participants | Risk level/comorbidities | Outcome(s) | Quality |
|---|---|---|---|---|---|---|
| Lipkus et al,[38] 2001 | Breast | 0 | 121 members of general public recruited through newspaper advertisements | Mean 10-year risk 2.65% (SD 1.13) | Mammography screening and intentions | M |
| Rimer et al,[40] 2002 | Breast | 1 and 2 years | 752 women aged 40–44 and 50–54 years enrolled in a personal care plan | Mean 10-year risk 2.7% | Mammography | M |
| Rubinstein et al,[39] 2011 | Breast, ovarian, colon | 6 months | 3786 patients from primary care clinic records with no history of colon, breast or ovarian cancer invited by mail following record review | 34% moderate or strong risk of ≥1 of the cancers | CRC screening, mammography | M |
| Schroy et al,[36] 2011 | Colorectal | 0 | 666 patients due for bowel screening identified from monthly audits of one hospital's electronic medical record | Average | Preferences, satisfaction with the decision-making process, screening intentions and test concordance | M–H |
| Schroy et al,[32] 2012 | Colorectal | 0, 1, 3, 6 and 12 months | 825 patients due for bowel screening identified from monthly audits of one hospital's electronic medical record | Average | Completion of a CRC screening test | H |
| Seitz et al,[37] 2016 | Breast | 0 | 2918 women aged 35–49 years with no history of breast cancer or a genetic mutation in BCRA1 or BCRA2 recruited through a survey company | 42% 10-year risk <1.5% (mean 1.08 SD 0.01); 58% 10-year risk ≥1.5% (mean 2.53 SD 0.04) | Mammography intentions | M |
| Sequist et al,[43] 2011 | Colorectal | 1 and 4 months | 1103 patients from 14 ambulatory health centres who were overdue for CRC screening | Average | CRC screening | M |
| Sherratt et al,[23] 2016 | Lung | 6 months | 297 current and 216 recent former smokers aged 18–60 years without a history of lung cancer and attending smoking cessation services | Not given | Smoking status | H |
| Trevena et al,[33] 2008 | Colorectal | 1 month | 314 patients recruited from six primary care practices without a history of CRC | Not given | Screening intentions, CRC screening | M |

BCRA, Breast Cancer Risk Assessment; CRC, colorectal cancer; FOBT, faecal occult blood test; H, high; L, low; M, medium.

**Table 2** Details of the risk-based interventions in each of the included studies

| Author, year | Risk tool | Intervention group(s) | Comparison (where applicable) | Format of risk |
|---|---|---|---|---|
| Bodurtha et al,[31] 2009 | Gail model (5 year and lifetime) | Information sheets with risk level and handouts addressing traditional constructs of Health Belief Model including barriers to mammography, breast cancer seriousness, individual risk for breast cancer and benefits of yearly mammography | General information about breast cancer prevention practices, including mammography | Usual (<15%), moderate (15%–30%) or strong (>30%) |
| Bowen et al,[41] 2006 | Gail model (5 year, 10 year and at age 79) | Four weekly 2-hour sessions led by a health counsellor focusing on risk assessment and education, screening, stress management and social support | Delayed intervention | No details given |
| Bowen and Powers,[42] 2010 | Gail model (lifetime) | Information sheets with general information on breast cancer risk and personalised risk information plus telephone counselling and offer for more intensive group or genetic counselling | Delayed intervention | Bar graph of absolute lifetime risk along with age-appropriate estimates for the 'average risk' woman |
| Davis et al,[34] 2004 | BCRA tool (updated version of Gail model) (lifetime) | 10 min brief intervention designed to increase accuracy of perceived risk including results of risk assessment and screening recommendations tailored to participant's stage of adoption of mammography and follow-up written information | No intervention | Verbal over the telephone. No additional details given |
| Glanz et al,[45] 2013 | Children's BRAT | Three mailings with personalised risk feedback, interactive skin cancer education materials, a family fun guide and suggestions for overcoming barriers and reminders to engage in preventive practices | Single mailing of standardised skin cancer information | No details given |
| Glazebrook et al,[22] 2006 | No details given | Self-directed computer program including sections on skin protection, how to detect melanoma, dangers of sun exposure, how to check skin, how to reduce risk and individualised feedback of risk | Usual care | Comparative risk |
| Greene and Brinn,[44] 2003 | Relative risk adapted from 'ADD Wants to Convert' | Self-assessment of risk alongside generic messages about tanning, tanning beds and sun exposure | Generic messages about tanning, tanning beds and sun exposure | Numerical scale from 1 to 36 |
| Helmes et al,[35] 2006 | Gail model (lifetime) | Face-to-face or telephone intervention consisting of 8 items: (1) a personal risk sheet, (2) a personal computer-drawn pedigree, (3) a 23-page participant booklet, (4) breast self-examination (BSE) brochure, (5) pap smear and mammography brochure, (6) BSE shower card, (7) pictures of chromosomes and gene mutations and (8) a list of community resources for breast cancer | No intervention | Bar charts of absolute % risk with numerical % alongside for the individual, an average-risk woman and a high-risk woman |

**Table 2** Continued

| Author, year | Risk tool | Intervention group(s) | Comparison (where applicable) | Format of risk |
|---|---|---|---|---|
| Holloway et al,[21] 2003 | Wilkinson score | Brief 10min counselling session integrated with smear test appointment including relative and absolute risks and then negotiation of appropriate screening intervals | Usual care | Comparative and absolute risk in pictures and numbers |
| Lipkus and Klein,[30] 2006 | Not given | Written information about CRC, CRC screening methods and CRC risk factors plus either (1) tailored CRC risk factor information or (2) tailored CRC risk factor information plus information on whether their total number of CRC risk factors was greater or not than average | Written information about CRC, CRC screening methods and CRC risk factors | Narrative comparative risk |
| Lipkus et al,[38] 2001 | Gail model (10 year) | One-page handout describing the Gail model plus absolute risk alone | As for intervention group plus how their risk compared with a woman of their age and race at the lowest level of risk | Absolute risk±risk of a woman at the lowest level of risk as percentages in a pie chart |
| Rimer et al,[40] 2002 | Gail model (10 year and lifetime) | Tailored print booklet and brief tailored newspaper plus personalised risk | Usual care (postcard reminder) | Absolute risk as a percentage |
| Rubinstein et al,[39] 2012 | Family Healthware tool | Written personalised risk assessment and tailored prevention messages | Written generalised prevention messages | Qualitative risk—weak, moderate or strong familial risk |
| Schroy et al,[36] 2011 | Harvard cancer risk model (10 year) | Interactive 20–30min computer-based decision aid plus personalised risk assessment | Interactive 20–30min computer-based decision aid alone | Thermograph, indicating where the participant is along with a description, for example, your risk is below average |
| Schroy et al,[32] 2012 | Harvard cancer risk model (10 year) | Interactive 20–30min computer-based decision aid plus personalised risk assessment followed immediately by a meeting with their providers to discuss screening and identify a preferred screening strategy. Providers received written notification hand-delivered by all the patients acknowledging that they were participating in the 'CRC decision aid study' at the time of the visit to ensure that screening was discussed | As for intervention but without personalised risk assessment | Qualitative framing ('very much below average risk' to 'very much above average risk') with accompanying suggestions for behaviour modifications that might reduce risk, including a strong recommendation for screening, regardless of risk |
| Seitz et al,[37] 2016 | Gail model (10 year) | Online risk plus basic information about mammography and national recommendations plus either (1) statements about women making choices, (2) untailored examples of women making choices or (3) examples of similar women making choices | No information or the same basic information as intervention group | Absolute risk and risk of an average risk age-matched women as numeric frequencies and icon arrays |

Continued

## Table 2 Continued

| Author, year | Risk tool | Intervention group(s) | Comparison (where applicable) | Format of risk |
|---|---|---|---|---|
| Sequist et al,[43] 2011 | Harvard cancer risk model (10 year) | Personalised electronic message highlighting their overdue screening status and providing a link to a web-based tool to assess their risk | No contact | Comparative risk on seven-point ordinal scale from very much below average to very much above average and in interactive graphical format |
| Sherratt et al,[23] 2016 | Liverpool Lung Project model (5 year at age 70) | Personalised risk plus booklet stating the association between smoking and lung cancer and highlighting that quitting smoking was the best thing to do | As for intervention but without personalised risk assessment | Verbal and written absolute risk if continue to smoke and if stop smoking alongside icon arrays |
| Trevena et al,[33] 2008 | No details given | 20-page booklet including personalised risk, absolute reduction in CRC mortality with screening over the next 10 years, probability of test outcomes from screening and information about how to get screened. | Three-page booklet with information and recommendations about screening | Words and 1000-face diagrams |

AAD, American Association of Dermatology; BCRA, Breast Cancer Risk Assessment; BRAT, Brief Skin Cancer Risk Assessment Tool; CRC, colorectal cancer.

Other risk models used were the Liverpool Lung Project model,[26] Family Healthware tool,[27] Wilkinson score for cervical cancer[28] and the Brief Skin Cancer Risk Assessment Tool[29] adapted for children. Quality assessment for each of the study is provided in online supplementary file 2. Seven were assessed as high or medium/high quality, 11 as medium quality and one as medium/low.

Overall findings and evidence synthesis along with the number and quality of studies addressing each outcome are summarised in table 3.

### Preferences and intentions for screening
#### Preferences for screening

Two RCTs reported participants' views about screening. In the cluster-randomised trial by Holloway et al,[21] participants who received a 10 min counselling session including information about relative and absolute risks of cervical cancer integrated within a smear test appointment were significantly less likely to state a preference for the next interval for cervical screening to be 12 months or less than those who received usual care (OR 0.51 (95% CI 0.41 to 0.64)). The second study by Lipkus and Klein[30] reported attitudinal ambivalence towards faecal occult blood test (FOBT) screening measured by their agreement with three Likert-style items stating that they had 'mixed feelings', felt 'torn' and had 'conflicting thoughts' about whether to get screened for CRC using an FOBT. Participants who received personalised estimates of either absolute or absolute plus comparative risk alongside written information about CRC screening had significantly lower ambivalence than those who received the same written information without tailored CRC risk information (P<0.05).

#### Intention to attend cancer screening

Eight studies assessed intentions to attend cancer screening: five for mammography and three for CRC screening. Five showed no effect of risk information, three in which the only substantial difference between the intervention and control groups was the provision of a risk estimate.[31–33] Bodurtha et al[31] found no significant differences at 18 months between those randomised to receive either printed sheets with their 5-year and lifetime estimates of breast cancer risk alongside information addressing barriers to mammography, breast cancer seriousness and benefits of yearly mammography or general information about breast cancer prevention practices not tailored to their risk level (OR after adjusting for baseline intentions and recruitment site 0.97 (95% CI 0.70 to 1.33)). Davis et al[34] reported that women who received a brief intervention over the telephone including information about lifetime risk of cancer and screening recommendations were no more likely at 1 month to report being in the maintenance stage (having had one mammogram in the past 2 years and two or more in the past 4 years and planning to get another on schedule) than the control group who received no intervention (67% in the intervention group compared with 68% in the control group). Helmes et al[35] reported changes in a single breast health intentions

**Table 3** Summary of evidence on outcomes

| Outcome measure | No of studies | Studies with significant positive effect | Studies with no effect | Best evidence synthesis |
|---|---|---|---|---|
| **Screening** | | | | |
| Preferences for screening | 2 | One medium-quality/high-quality and one high-quality RCT | None | Evidence of positive effect |
| Intention to attend screening | 8 | One medium-quality RCT* | One high-quality, one medium-quality/high-quality and four medium-quality RCTs* | Evidence of no effect |
| Attendance at screening | 12 | One high-quality RCT | Two high-quality, two medium-quality/high-quality and seven medium-quality studies | Evidence of no effect |
| **Health-related behaviours** | | | | |
| Intention to change health-related behaviours | | | | |
| To tan | 1 | One low/medium RCT | None | Limited evidence of positive effect |
| To protect skin | 1 | None | One low/medium RCT | Limited evidence of no effect |
| Health-related behaviours | | | | |
| Smoking cessation | 1 | None | One high-quality RCT | Limited evidence of no effect |
| Smoking abstinence | 1 | One high-quality RCT | None | Limited evidence of positive effect |
| Sun protection | 2 | Two medium-quality RCTs | | Indicative evidence of positive effect |
| Tanning bed usage | 1 | None | One low/medium RCT | Limited evidence |
| Adult skin examination | 2 | Two medium-quality RCTs | None | Indicative evidence of positive effect |
| Child skin examination | 1 | None | One medium-quality RCT | Limited evidence of no effect |
| Breast examination | 3 | Two high-quality RCTs | One medium/high RCT | Indicative evidence of positive effect |
| Diet | 0 | None | None | No evidence |
| Physical activity | 0 | None | None | No evidence |
| Alcohol | 0 | None | None | No evidence |

*One medium-quality study reported a significant positive effect in low-risk women and no effect in high-risk women.
RCT, randomised controlled trial.

measure which included intention to have mammography, clinical breast examination and breast self-examination. They found no significant differences at baseline (P=0.23) or 3-month follow-up (P=0.46) between women who received estimates of their lifetime risk of breast cancer along with information about breast awareness either face-to-face or over the telephone and a control group who received no intervention. Schroy et al[36] randomised participants to complete an interactive 20–30 min computer-based decision aid which either did or did not include a personalised risk assessment. There was no difference between groups on a five-point scale of how sure they were that they would schedule a CRC screening test (mean scores 4.3 (SD 1.0)

for both groups). Trevena et al[33] similarly reported no effect on intention to have CRC screening of a 20-page decision aid including information about baseline risk and absolute reduction in CRC mortality with screening, compared with a three-page booklet with information and recommendations about screening.

The two studies reporting an effect were by Lipkus and Klein[30] and Seitz et al.[37] In Lipkus and Klein, intention to complete an FOBT that would be given to them within the following month was measured on a seven-point Likert scale. The intentions reported by participants who received absolute risk (mean 3.65, n=40) or absolute plus either low (mean 6.43, n=38) or high (mean

6.65, n=39) comparative risk information were statistically significantly higher (P<0.05) than those participants in the control group who were provided with the same written information but without risk estimates (mean 2.21, n=43). The mean intention reported by the group which received the comparative risk was also significantly higher than for the absolute risk only group. In Seitz *et al*, women were separated into those with an estimated 10-year breast cancer risk above or below 1.5%. Intention to wait until age 50 before undergoing a mammogram was measured for those with a risk <1.5% and intention to start or continue to undergo mammograms in their 40s for those with a risk ≥1.5. In the low-risk group, all risk-based intervention conditions resulted in a significant increase in the percentage of women planning to wait to age 50. However, in the high-risk group, no such significant difference was seen.

The eighth study by Lipkus *et al*[38] reported the difference in intentions to get a mammogram between one group that received a one-page handout including their estimated absolute risk and another group that received the same handout plus information concerning how their risk compared with a woman of their age and race at the lowest level of risk. Immediately after the provision of risk information, overall 2.5%, 67.8% and 24.8% reported that the risk information lowered, did not affect or increased their intentions to undergo a mammogram, respectively, with no differences between the groups.

### Attendance at screening

Twelve RCTs reported attendance at screening: six for mammography[31 34 39–42], five for CRC[30 32 33 39 43] and one

for cervical cancer.[21] Except for one high-quality RCT in which the intervention group received information sheets including general information on breast cancer risk alongside personalised risk information and telephone counselling and the offer for more intensive group or genetic counselling,[42] all showed no effect of the risk-based interventions as shown in the meta-analysis (figure 2) with a combined RR of 1.02 (95% CI 0.98 to 1.03, I$^2$ 61.6%).

### Intention to change health-related behaviours
#### Intention to tan or protect skin

One RCT by Greene and Brinn measured intention to tan on a six-item Likert-type scale and intention to protect skin using a three-item scale.[44] Participants who completed a self-assessment risk score alongside receiving generic information about tanning, tanning beds and sun exposure reported significantly decreased intentions to use tanning beds than those receiving the same generic information alone (2.68, n=70 compared with 3.19, n=71, P<0.05). In contrast, there were no significant differences in intentions to protect skin (2.38, n=70 compared with 2.49, n=71, P>0.05).

### Change in health-related behaviours
#### Smoking status

One high-quality RCT[23] reported the impact of risk information on smoking status. Receiving a personalised risk estimate in addition to a generic leaflet did not predict self-reported smoking status at 6 months in current smokers (P=0.66) but was associated with an increased odds of remaining a former smoker in those who had recently quit (OR 1.91 (95% CI 1.03 to 3.55)).

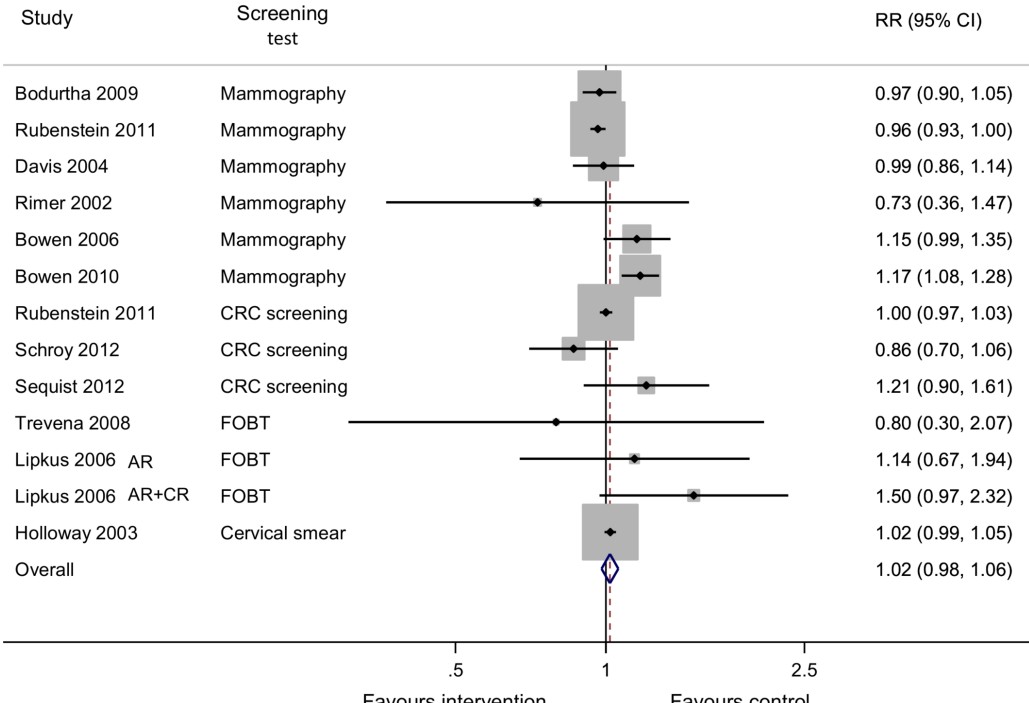

**Figure 2** Relative risk for adherence to recommended screening postintervention. AR, absolute risk; CR, comparative risk; CRC, colorectal cancer; FOBT, faecal occult blood test.

## Sun exposure and sun protection habits

Two RCTs[22 45] measured sun protection habits by survey completion at baseline and follow-up. One by Glanz *et al* compared the effect on childhood sun exposure and sun protection habits of three mailings with personalised risk feedback, interactive skin cancer education materials and a family fun guide to a single mailing of standardised skin cancer information.[45] The other by Glazebrook *et al* compared usual care with a self-directed computer program including individualised feedback of risk alongside sections on skin protection, how to detect melanoma, dangers of sun exposure, how to check skin and how to reduce risk.[22] Both showed increases in overall sun protection habits (increase in sun protection habits index 0.19 in the intervention group compared with 0.14, P=0.02)[45] and mean difference in skin protective behaviour score between intervention and control at 6-month follow-up 0.33 (95% CI 0.09 to 0.57)[22] with variable results for individual aspects including wearing a sun hat, wearing a shirt, wearing sunglasses, use of sun cream, number of sunburns, staying in the shade and sun exposure during weekdays and weekends.

## Tanning bed usage

The RCT by Greene and Brinn[44] measured change in tanning behaviour and tanning bed usage. Participants who completed the self-assessment risk score reported lower rates of tanning bed usage in the previous month at follow-up (2.18, n=70 compared with 3.76, n=71, P<0.05) but no difference in change in tanning behaviour from preintervention to postintervention (−1.25, n=70 compared with −2.08, n=71, P>0.05).

## Self/parent skin examination

The two RCTs by Glanz *et al* and Glazebrook *et al*,[22] measured rates of skin examination in adults or parents and children.[45] Both showed statistically significant increases among adults and parents receiving personalised risk information (P<0.05), whereas the increase in parents examining their children was not statistically significant (P=0.06).

## Clinical breast examination and breast self-examination

Three RCTs[31 41 42] measured rates of clinical breast examination and/or breast self-examination following provision of risk information. In the RCT by Bodurtha *et al*, no significant differences were seen between those randomised to receive printed sheets including estimates of 5-year and lifetime risk of breast cancer alongside information addressing barriers to mammography, breast cancer seriousness and benefits of yearly mammography and those receiving general information about breast cancer prevention practices not tailored to their risk level for either frequency of clinical breast examination (crude rates: 91.4% vs 91.0%; adjusted OR 1.00 (95% CI 0.60 to 1.66)) or breast self-examination (crude rates: 56.8% vs 57.6%; adjusted OR 0.95 (95% CI 0.67 to 1.33)).[31] The other two studies, both by Bowen *et al*, found significantly

(P<0.01) greater increases in the proportion reporting performing breast self-examination in the intervention groups (35% to 52% and 36% to 62%) compared with controls (33% to 36% and 38% to 40%).[41 42] However, both these studies compared intensive interventions (four weekly 2-hour sessions led by a health counsellor[41] or information sheets plus telephone counselling and the offer of more intensive group or genetic counselling[42]) with delayed intervention.

## DISCUSSION

This systematic review is, to our knowledge, the first review of the impact of interventions delivered across multiple settings which incorporate personalised information about cancer risk on intention to change health-related behaviour and health-related behaviours themselves in the general population. The findings show that such interventions do not affect intention to attend or attendance at screening. There is limited evidence that they increase smoking abstinence, sun protection, adult skin self-examination and breast examination and decrease intention to tan. However, this was not seen for smoking cessation, parental child skin examination or intention to protect skin. There is a notable absence of studies assessing the impact on diet, physical activity and alcohol consumption with only one reporting smoking status and none including objective measures of behaviour.

Our finding that interventions incorporating personalised information about cancer risk had no effect on intention to attend or attendance at screening is consistent with a previous Cochrane review in which personalised risk communication had little effect on the uptake of screening tests (fixed-effect OR 0.95 (95% CI 0.78 to 1.15)).[16] However, as in that review, there was evidence of increased concordance between screening preferences and recommendations and decreased ambivalence. This supports the suggestion made in that review that personalised risk information might be useful for shared and informed decision-making. For example, in surveys of participants about their knowledge and values for cancer screening decisions and decision-making processes, only 21% report feeling extremely well informed,[46] and the majority overestimate lifetime risk of cancer incidence and mortality.[46 47] While providing individuals with information about their estimated cancer risk may therefore not influence overall rates of screening, it may contribute to the decision to take up screening or not at an individual level and support shared decision-making.

The absence of significant effects on health-related behaviours is also consistent with research in other disease areas, such as cardiovascular disease, where systematic reviews have found only few studies reporting behaviour change and no significant effects on lifestyle.[48–50] This is perhaps not surprising given that behaviour change is influenced by many other factors, including health beliefs, social context, the environment and personal attributes such as time orientation.[12 13] However, there was

no evidence that interventions that include information about cancer risk result in harm through false reassurance and the adoption of unhealthy behaviours. This is important as on average many of the general population overestimate their own risk of cancer,[30][35][41][51–53] and so if information about cancer risk were routinely provided within clinical practice, large numbers would be receiving an estimate lower than their prior perceptions.

The main strengths of this review are the systematic search of multiple electronic databases and the broad inclusion criteria. This allowed us to include studies that assess the impact of interventions incorporating personalised cancer risk information on multiple behavioural outcomes. However, from nearly 40 000 titles and abstracts, we only included 14 with an additional five found through citation searching. This highlights the challenge in identifying studies in this area in which the primary purpose may not be related to the provision of personalised risk information. There was also significant heterogeneity in the outcome measures included, duration of follow-up and method of recruitment across the included studies. For all outcomes, except attendance at screening, there were either too few studies to meaningfully pool results or each study used different non-comparable measures. Even for attendance at screening for which meta-analysis was possible, we were only able to pool crude estimates, and the included studies addressed screening for breast, bowel and cervical cancer. While it is possible that the impact on screening attendance might be different across the different cancer sites because of the nature of the tests involved, the finding that only one study of mammography showed an effect of interventions incorporating personalised cancer risk information suggests that this is unlikely to be the case. The duration of follow-up also varied from 1 to 18 months. However, the studies with shorter follow-up were those with intention as the outcome measures and, of the 10 studies reporting health-related behaviours, five had a follow-up period of a year or more and three a period of 6 months. It is therefore unlikely that the studies as a whole were too short to detect changes in behaviour or reflected only immediate unsustained changes.

A further limitation is that many of the interventions consisted of provision of personalised risk information alongside a range of additional information, either written or delivered in person or in groups. Separating the effect of the risk information from those additional elements of the interventions was therefore not possible. However, we chose not to exclude these studies from this review because it is unlikely that personalised risk information would be incorporated into routine practice in isolation and, if anything, including them would overestimate the effect of the personalised risk information. It is also possible that the findings do not reflect the potential impact of interventions incorporating personalised information about cancer risk on the general population as a whole: half of the included studies focused on female cancers and so only recruited women and all were subject to recruitment bias with the participants who agreed to take part potentially more interested in their cancer risk or more healthy, resulting in a bias in either direction.

In addition to these specific limitations of our review, the findings also suggest a number of areas for future research. In particular, the absence of studies assessing the impact on diet, physical activity and alcohol consumption and only one study reporting smoking cessation demonstrate the need for trials assessing change in these behaviours, preferably measured objectively, including measures of other theory-based determinants of behaviour change (eg, self-efficacy). Only with such data will we be able to assess whether such individualised approaches have a place alongside population-wide prevention strategies.

**Acknowledgements** The authors thank Isla Kuhn, Reader Services Librarian, University of Cambridge Medical Library for her help in developing the search strategy.

**Contributors** JAU-S developed the protocol, completed the search, screened articles for inclusion, extracted data, synthesised the findings, interpreted the results and drafted the manuscript. BS developed the protocol, screened articles for inclusion, extracted data, interpreted the results and critically revised the manuscript. SJS synthesised the findings and critically revised the manuscript. KM extracted data, interpreted the results and critically revised the manuscript. SJG developed the protocol, screened articles for inclusion, interpreted the results and critically revised the manuscript. All authors approved the final version.

**Funding** JAU-S and KM are funded by a Cancer Research UK/BUPA Foundation Cancer Prevention Fellowship (C55650/A21464). BS was supported by the Medical Research Council (MC_UU_12015/4). SJS is supported by the Medical Research Council www.mrc.ac.uk (Unit Programme no MC_UU_12015/1). The University of Cambridge has received salary support in respect of SJG from the NHS in the East of England through the Clinical Academic Reserve.

**Competing interests** None declared.

**Patient consent** Not required.

**Provenance and peer review** Not commissioned; externally peer reviewed.

**Data sharing statement** All data are available from the reports or authors of the primary research. No additional data are available.

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
