## [Reviewer comments · BMJ Open]

ARTICLE DETAILS

TITLE (PROVISIONAL)	Effect of interventions incorporating personalised cancer risk information on intentions and behaviour: a systematic review and meta-analysis of randomised controlled trials
AUTHORS	Usher-Smith, Juliet; Silarova, Barbora; Sharp, Stephen; Mills, Katie; Griffin, Simon

VERSION 1 – REVIEW

REVIEWER	Erika Waters Washington University in St. Louis, USA Part of my research program is related to developing and testing risk prediction models.
REVIEW RETURNED	31-May-2017

GENERAL COMMENTS	Major Concerns The most significant concern is that the manuscript seems to be completely divorced from the health behavior theory literature. This is a major limitation because perceived risk is mentioned in some form (e.g., perceived threat, perceived susceptibility, perceived vulnerability, perceived likelihood, perceived risk) in nearly every theory of health behavior and volitional behavior change published. Although the health behavior theories state that perceived risk is important, they also state that (a) other health beliefs are also important, such as self-efficacy, and (b) although the link between behavioral intentions and behavior change is modest, it can be bolstered by intervention that provide other sorts of support for individuals who want to change. In fact, several stage theories of behavior change (e.g., Precaution Adoption Process Model, Transtheoretical Model, Health Action Process Approach) explicitly state that different intervention components are needed to (a) alert people of a problem that they need to attend to, (b) motivate them to act, (c) help them act, and (d) maintain their action. Other theories, such as the social ecological model, account for the social context/environment by highlighting the fact that behavior is also influenced by factors outside an individual's control, such as availability of medical specialists, physician recommendation, the size of any co-pays, and whether or not one has access to sick leave (which, in the U.S., is not guaranteed even for doctor's appointments). By not grounding the study or results in the health behavior theory literature, the authors risk oversimplifying the process of behavior change and overstating their conclusions.
--

	In short, is it reasonable to expect an intervention that provides 1 piece of information at 1 point in time to produce lasting change in activities as complex and multi-determined as cancer prevention and detection behaviors? Something else to be considered is the potential value personalized risk tools might have for fostering informed and/or shared medical decision making (see work by Stacy Sheridan or Michael Pignone). The other major concern is related to the results presented in the psychological well-being section. Whereas the findings related to risk perception and accuracy were correctly described in the context of participants' risk status, the results for psychological well-being were not. For example, the tools are described as reducing worry over time -- was this only for people who initially overestimated their risk (i.e., they were relieved at a "good" result) or was this for people who initially overestimated and underestimated their risk? Or was this information not provided in the analyses provided by the original authors? If the latter, it should be noted as a potential area for future research. The same issue is relevant for the anxiety and depression sections. It was surprising to see the SF-36 as an outcome, since it is typically not used in health prevention/promotion settings. What is the conceptual justification for including it? Minor concerns P. 9, line 29: Please provide an explanation as to why the 7 studies could not be included in the meta-analysis. P. 18, lines 18-31: Please either provide a citation supporting the assertion that provision of "low risk" information might encourage the adoption of unhealthy behaviors or indicate the speculative nature of the statement. P. 18, lines 48-52: Please provide a citation for the assertion that comparative risk is a more emotive construct and also a citation for the idea that it may be more vulnerable to bias and resistant to change. P. 19: The Cochran review cited also proposes that personalized risk information might be utilized for shared and informed decision making, since screening uptake itself is often dependent upon other factors. Please clarify this in the 1st paragraph on this page.
--	--

REVIEWER	Zhen Wang Mayo Clinic, USA
REVIEW RETURNED	01-Jul-2017

GENERAL COMMENTS	This study is a systematic review and meta-analyses of cancer risk based interventions. A total of 32 studies were included in the analyses. While the process of conducting systematic review was adequate and presented well, I am quite confused about the rationale and presentation of the findings. Specifically, I have the following suggestions and comments.
---

	1. The systematic review focused on interventions that provided a personal estimate of future cancer risk. It's safe to say that risk assessment of study participants after the interventions will improve if they were given a scientific estimate. Improvement doesn't mean a negative or positive value on risk assessment, which really depends on baseline evaluation on average from the study participants (overestimate or underestimate risk). So I feel that pooled effect size for absolute/comparative risk doesn't mean anything unless the authors can separate studies to underestimate/overestimate from the baseline. 2. The interventions are so different from the 32 studies. All of the participants received a package of interventions, including risk estimates. The current manuscript only focused on the risk estimate part, without much information about other components and no details about those tools used to estimate risk. Feels like comparing apples to oranges. The authors may add paragraphs to summarize other components of the interventions in the literature and add rationales for combining these studies are valid. 3. I am not sure why SMD (standardized mean difference) was used to pool perceived absolute risk or comparative risk. Risk is unit-free. Weighted mean difference should be good. Please describe. Also please be consistent with SMD. Sometimes standardized difference in means was used while in other places, standardized mean difference was used. 4. Abstract, please add description of perceived absolute risk and comparative risk and move the description of these terms in the methods. Also a summarized definition of all outcomes is necessary. 5. Please clarify the difference for two sets of outcomes (perceived absolute/comparative risk vs. absolute/comparative risk (without perceived)). And please describe how studies measure absolute/comparative risk. 6. Why used "comparative risk"? "relative risk" is more common? 7. Introduction, please add "US" to the NCI. Not sure if other countries have NCI. 8. Page 5 line 47, pre-post intervention studies are also observational studies. Please correct those in the exclusion (page 6 line 5). 9. Please add text to describe items in the quality assessment. 10. Page 10, line 40-44. Please describe the definition of "more accurate". The authors had to dichotomize the outcomes to get pooled RR. Why not use RR in "perceived risk" page 8-9?
--	--

REVIEWER	Keeble University of Leeds, UK
REVIEW RETURNED	07-Jul-2017

GENERAL COMMENTS	This clearly explained review considers the impact of cancer risk information interventions in the adult population. General comments:  -It is unfortunate that the pooling of results was limited, causing much of the article to be descriptive. Can the authors comment further on the data available for use in a meta-analysis? -Could the authors comment on the selection of the search terms used? There were over 30,000 articles returned, yet only 26 of these were included in the synthesis. An additional 6 were included following a handsearch outside of the initial 30,000. -The review would benefit from further details of how the meta-analysis was conducted, with references for the method(s) and any Stata packages used. -A discussion of potential biases in the articles used would be beneficial, such as the types of people recruited. For example, were participants recruited at random from the population, or were they volunteers who could potentially be more likely to worry about their cancer risk? -Could the authors comment further on the moral obligation not to reduce worry, from a medical rather than psychological perspective. Minor comments:  - Hyphens may be useful in the title and introductory paragraphs (eg, risk-based). -Many of the terms used (such as 'worry') may be subjective, can the authors comment on the potential weaknesses of subjective measures? -What role may the length of follow-up time in the studies have played? Could some studies be affected by short-term behavioral changes? -Several of the cancers in the 32 articles were predominantly female cancers. Could gender have played a role in the findings of this review?
---

REVIEWER	Eva Lorenz Institut für Medizinische Biometrie, Epidemiologie und Informatik 55101 Mainz
REVIEW RETURNED	10-Jul-2017

GENERAL COMMENTS	General comments I was pleased to review this systematic review on the impact of cancer risk based interventions. I believe it is an important topic and will be of interest to many readers. This systematic review and meta analysis has many strengths including a thorough search of literature, the different types of cancer and thereby interventions, and a variety of outcomes. One major methodology concern is the combination of diverse risk formats of the same studies in the pooled analyses. Since studies seem to be rather heterogeneous in the reporting of risk measures, the authors might remove or redesign the meta-analyses on 'risk perception and understanding of risk estimate' and on 'psychological well-being'. I would, in general, recommend removing the meta-analyses as the study heterogeneity between studies was rather high and the number of distinct studies in the separate analyses was rather low, e.g. only one study was considered for the pooled estimate of perceived comparative risk which is methodologically not meaningful. Title Please elaborate on the term impact. What is actually the outcome of interest in this study? Abstract I 13: Please specify the reason why the search started in 2000 and not earlier. This can either be done here or on page 5 in the section Search strategy. There is no information on study heterogeneity given. Can you please add a sentence on the heterogeneity of studies (e.g. design, size) and outcomes? I. 31: Please introduce the abbreviation 'RR' before first use. I. 36: Please harmonize the spelling of 'risk based' 'risk-based' throughout the manuscript (e.g. spelling in title differs from abstract). Box with strengths and limitations: Please add: 'This systematic review and meta-analysis is the first...' The study heterogeneity is listed in this box but should already be mentioned in the abstract. How did the heterogeneity affect the pooling of results? Please specify. Introduction p. 5, l. 5: Please introduce RCT as an abbreviation for randomized controlled trial.
--

The research question is not specifically defined. What do you mean by 'impact of the provision of cancer risk-based interventions'. Impact on what? Which different types of outcomes do you intend to investigate?

Methods

p.6, l.16: How was SGs rating incorporated in the rating? Did JUS and BS both review all articles or did they split the articles and then a random sample was reviewed by a third reviewer? Please specify if the first two reviewers did not individually read each of all articles.

p.6, l.56: Please introduce the abbreviation for Critical Appraisal Skills Programme before first use.

p.7, l.38: Please remove: 'we were only able to do this for a small number of outcomes' to the results section.

p. 7, l. 47: Please write Stata instead of STATA.

p.8, l. 2: Same issue as previously: Did the first two reviewers share the screening or did they both read each article?

Please add numbers of articles to the exclusion reasons.

p.8, l.51: Consider adding percentages in order to know the relative proportion of 18 studies out of 32.

p.9, first paragraph: The 'format of the risk' is not introduced or described in detail. It is not clear to me as to why the authors report the estimates in e.g. the study by McCaul in 6 different groups (format of risk and time from intervention). Why are CR groups even included in the calculation of the perceived absolute risk? Each study should contribute once to the overall estimate for each format of risk category separately. Times from intervention are also not comparable and treating them equally should be avoided. If the difference post intervention and at baseline are to be pooled in the forest plot, each study should only occur once with one effect/difference estimate.

p.11, paragraph 'cancer worry': It seems as if you are again combining various effect estimates from overlapping study populations in one meta-analysis. Absolute and comparative risks cannot be treated as if they were the same and should be investigated in separate analyses. The intervention effects of the joint group AR+CR is always between the single AR and CR intervention effects. I assume this is true because the joint group includes the exact same individuals from both groups. If this is true, effect estimates cannot be seen as independent sources of information and cannot be combined in the analysis. The study by Helmes is reported twice with the exact same format of risk and time for intervention but with differing intervention effect. This seems to be a mistake.

Please replace the meta-analyses by analyses in subgroups where you specifically only compare AR or CR with each other but do not mix all different types in one pooled analysis since they are not comparable.

	p.15, paragraph 'attendance at screening': You again excluded an article from the meta-analysis because no significant effect was observed. No presence of an affect is, however, a relevant piece of information and must be included in the analysis. The reason for excluding six studies from the analysis because of the absence of significant intervention effects is methodologically not tolerable. The strength of the intervention effects does not represent the actual intervention effect but a likely overestimation of effects. Non- significant effect estimates need to be considered in the analyses as well. Figure 1: The PRISMA flow diagram should also report the number of articles considered for meta-analyses (by type of meta-analysis) and not only those considered in the review.
--	--

VERSION 1 – AUTHOR RESPONSE

Reviewer: 1

Reviewer Name: Erika Waters

Institution and Country: Washington University in St. Louis, USA Competing Interests: Part of my research program is related to developing and testing risk prediction models.

The manuscript "The impact of cancer risk based interventions to people at population level risk: a systematic review and meta-analysis is timely considering the increased emphasis in the U.S. on precision and personalized medicine initiatives. It is also likely of interest to BMJ Open readers. Overall the manuscript is well-written and has identified the vast majority of the eligible articles published between 2000 and 2015. However, it has several limitations that reduce my enthusiasm for publication.

We are pleased that the reviewer believes this manuscript is timely and likely of interest to BMJ Open readers. Below we address each of the limitations mentioned and believe these have improved the manuscript.

Major Concerns

The most significant concern is that the manuscript seems to be completely divorced from the health behavior theory literature. This is a major limitation because perceived risk is mentioned in some form (e.g., perceived threat, perceived susceptibility, perceived vulnerability, perceived likelihood, perceived risk) in nearly every theory of health behavior and volitional behavior change published. Although the health behavior theories state that perceived risk is important, they also state that (a) other health beliefs are also important, such as self-efficacy, and (b) although the link between behavioral intentions and behavior change is modest, it can be bolstered by intervention that provide other sorts of support for individuals who want to change. In fact, several stage theories of behavior change (e.g., Precaution Adoption Process Model, Transtheoretical Model, Health Action Process Approach) explicitly state that different intervention components are needed to (a) alert people of a problem that they need to attend to, (b) motivate them to act, (c) help them act, and (d) maintain their action. Other theories, such as the social ecological model, account for the social context/environment by highlighting the fact that behavior is also influenced by factors outside an individual's control, such as availability of medical specialists, physician recommendation, the size of any co-pays, and whether or not one has access to sick leave (which, in the U.S., is not guaranteed even for doctor's appointments).

By not grounding the study or results in the health behavior theory literature, the authors risk oversimplifying the process of behavior change and overstating their conclusions.

In short, is it reasonable to expect an intervention that provides 1 piece of information at 1 point in time to produce lasting change in activities as complex and multi-determined as cancer prevention and detection behaviors?

Response: We agree with the reviewer that it is unlikely that provision of risk information on its own will lead to behaviour change. This is, however, largely based on the evidence from other diseases e.g. cardiovascular disease and type 2 diabetes. Systematic reviews focusing on whether incorporating information about cancer risk impacts intention and behaviour are lacking. Furthermore, we believe that our review is timely as online risk calculation tools are becoming more and more available and there is the potential now to incorporate cancer risk scores into clinical practice in the same way as they are currently used in cardiovascular disease and diabetes. It is therefore important to have an evidence on whether provision of cancer risk information will have positive impact on intentions and behaviour or lead to potential harm through false reassurance before risk tools are introduced into routine practice. To reflect this in the manuscript we have added the following text to the Introduction and Discussion respectively:

“Most behaviour change theories suggest that perceived risk is important alongside other constructs such as self-efficacy, response efficacy in promoting behaviour change^{12,13}. Providing individuals with estimates of their risk of cancer alongside other behaviour change interventions may therefore help motivate behaviour change at an individual level.”

“The absence of significant effects on health-related behaviours is also consistent with research in other disease areas, such as cardiovascular disease, where systematic reviews have found only few studies reporting behaviour change and no significant effects on lifestyle^{47–49}. This is perhaps not surprising given that behaviour change is influenced by many other factors, including health beliefs, social context, the environment, and personal attributes such as time orientation^{12,13}. However, there was no evidence that interventions that include information about cancer risk result in harm through false reassurance and the adoption of unhealthy behaviours. This is important as on average many of the general population overestimate their own risk of cancer^{30,35,40,50–52} and so if information about cancer risk were routinely provided within clinical practice large numbers would be receiving an estimate lower than their prior perceptions.”

Comment: Something else to be considered is the potential value personalized risk tools might have for fostering informed and/or shared medical decision making (see work by Stacy Sheridan or Michael Pignone).

Response: We thank the reviewer for highlighting this additional potential value. We have included the following text with references to some of the work by Michael Pignone in the discussion to reflect this:

“Our finding that interventions incorporating information about cancer risk had no effect on intention to attend or attendance at screening is consistent with a previous Cochrane review in which personalised risk communication had little effect on the uptake of screening tests (fixed-effect OR 0.95 (95% CI 0.78 to 1.15))¹⁴. However, as in that review, there was evidence of increased concordance between screening preferences and recommendations. This supports the suggestion made in that review that personalised risk information might be useful for shared and informed decision making. For example, in surveys of participants about their knowledge and values for cancer screening decisions and decision-making processes, only 21% report feeling extremely well informed⁴³ and the majority overestimate lifetime risk of cancer incidence and mortality^{43,44}.

While providing individuals with information about their cancer risk may therefore not influence overall rates of screening it may contribute to the decision to take up screening or not at an individual level and support shared decision making.”

The other major concern is related to the results presented in the psychological well-being section. Whereas the findings related to risk perception and accuracy were correctly described in the context of participants' risk status, the results for psychological well-being were not. For example, the tools are described as reducing worry over time -- was this only for people who initially overestimated their risk (i.e., they were relieved at a "good" result) or was this for people who initially overestimated and underestimated their risk? Or was this information not provided in the analyses provided by the original authors? If the latter, it should be noted as a potential area for future research. The same issue is relevant for the anxiety and depression sections.

In focusing the review on the impact on behaviour change, we have now removed this psychological well-being section from the manuscript.

Comment: It was surprising to see the SF-36 as an outcome, since it is typically not used in health prevention/promotion settings. What is the conceptual justification for including it?

Response: In the previous version of this manuscript we included the SF-36 as it had been included as an outcome in several of the included studies. It is a measure of general health and includes sections on general health perceptions and mental health which could reasonably be affected by being provided with one's risk of cancer. As described above, however, we have now removed this section from the manuscript.

Minor concerns

P. 9, line 29: Please provide an explanation as to why the 7 studies could not be included in the meta-analysis.

Response: They were initially excluded because they did not provide sufficient data. They are now excluded from the manuscript as they no-longer meet the inclusion criteria.

P. 18, lines 18-31: Please either provide a citation supporting the assertion that provision of "low risk" information might encourage the adoption of unhealthy behaviors or indicate the speculative nature of the statement.

Response: We have removed this section from the manuscript.

P. 18, lines 48-52: Please provide a citation for the assertion that comparative risk is a more emotive construct and also a citation for the idea that it may be more vulnerable to bias and resistant to change.

Response: We have removed this section from the manuscript.

P. 19: The Cochran review cited also proposes that personalized risk information might be utilized for shared and informed decision making, since screening uptake itself is often dependent upon other factors. Please clarify this in the 1st paragraph on this page.

Response: As suggested we have amended the text in that paragraph to reflect the fact that the use of personalised risk information for shared and informed decision making was proposed in the Cochrane review. The text now reads:

“Our finding that interventions incorporating information about cancer risk had no effect on intention to attend or attendance at screening is consistent with a previous Cochrane review in which personalised risk communication had little effect on the uptake of screening tests (fixed-effect OR 0.95 (95% CI 0.78 to 1.15))¹⁴. However, as in that review, there was evidence of increased concordance between screening preferences and recommendations. This supports the suggestion made in that review that personalised risk information might be useful for shared and informed decision making.”

Reviewer: 2

Reviewer Name: Zhen Wang

Institution and Country: Mayo Clinic, USA Competing Interests: none declared

This study is a systematic review and meta-analyses of cancer risk based interventions. A total of 32 studies were included in the analyses. While the process of conducting systematic review was adequate and presented well, I am quite confused about the rationale and presentation of the findings. Specifically, I have the following suggestions and comments.

1. The systematic review focused on interventions that provided a personal estimate of future cancer risk. It's safe to say that risk assessment of study participants after the interventions will improve if they were given a scientific estimate. Improvement doesn't mean a negative or positive value on risk assessment, which really depends on baseline evaluation on average from the study participants (overestimate or underestimate risk). So I feel that pooled effect size for absolute/comparative risk doesn't mean anything unless the authors can separate studies to underestimate/overestimate from the baseline.

Response: We agree with the reviewer that risk accuracy is more meaningful than change in risk perception. In revising this manuscript we have, however, now focused on the impact on behaviour. This section of the results has therefore been removed.

2. The interventions are so different from the 32 studies. All of the participants received a package of interventions, including risk estimates. The current manuscript only focused on the risk estimate part, without much information about other components and no details about those tools used to estimate risk. Feels like comparing apples to oranges. The authors may add paragraphs to summarize other components of the interventions in the literature and add rationales for combining these studies are valid.

Response: We agree with the reviewer. In this revised version we have included additional details about the tools used to estimate the risk at the beginning of the results section as below:

“Further details of the risk models used to calculate the risk estimate provided to participants and the format of the intervention(s) are given in Table 2. All eight studies providing information about breast cancer risk used the Gail risk model²². This was the first risk model developed for breast cancer and includes age, age at menarche, age at first live birth, number of previous biopsies, number of biopsies showing atypical hyperplasia, and number of first-degree relatives with breast cancer.

Where details were given (n=3), all studies on colorectal cancer used the Harvard Cancer Risk tool²³ which includes family history, height and weight, alcohol consumption, vegetable and red meat consumption, physical activity, screening history, a history of inflammatory bowel disease, and use of aspirin, folate and female hormones. Other risk models used were the Liverpool Lung Project

model²⁴, Family Healthware tool²⁵, Wilkinson score for cervical cancer²⁶ and the brief skin cancer risk assessment tool (BRAT)²⁷ adapted for children.”

In addition to the descriptions of the interventions in Table 2, we have also included details of the additional components the participants received throughout the results section when describing the results. For example:

“Three RCTs^{30,39,40} measured rates of clinical breast examination and/or breast self-examination after risk information. In the RCT by Bodurtha et al., no significant differences were seen between those randomised to receive printed sheets with their 5-year and lifetime estimates of breast cancer risk alongside information addressing barriers to mammography, breast cancer seriousness and benefits of yearly mammography and those receiving general information about breast cancer prevention practices not tailored to their risk level for either frequency of clinical breast examination (crude rates: 91.4% vs 91.0%; adjusted OR: 1.00 (95%CI: 0.60 to 1.66)) or breast self-examination (crude rates: 56.8% vs 57.6%; adjusted OR: 0.95 (95%CI: 0.67 to 1.33)³⁰. The other two studies, both by Bowen et al., found significantly ($p < 0.01$) greater increases in the proportion reporting performing breast self-examination in the intervention groups (35% to 52% and 36% to 62%) compared with controls (33% to 36% and 38% to 40%)^{39,40}. Both these studies, however, compared intensive interventions (four weekly 2-hour sessions led by a health counsellor³⁹ or information sheets plus telephone counselling and the offer of more intensive group or genetic counselling⁴⁰) with delayed intervention.”

“Two RCTs reported participants’ views about screening. In a cluster-randomised trial by Holloway et al.²⁰ participants who received a 10 minute counselling session integrated within a smear test appointment which included relative and absolute risks of cervical cancer were significantly less likely to state a preference for the next screening interval for cervical screening to be 12 months or less than those who received usual care (OR: 0.51 (95%CI: 0.41-0.64)). The second study by Lipkus et al.²⁹ reported attitudinal ambivalence towards faecal occult blood test (FOBT) screening measured by their agreement with three Likert-style items stating that they had “mixed feelings”, felt “torn” and had “conflicting thoughts” about whether to get screened for CRC using an FOBT. Participants who received either absolute or absolute plus comparative risk alongside written information and CRC and CRC screening had significantly lower ambivalence than those who received the same written information without tailored CRC risk factor information ($p < 0.05$).“

We have also added the following text to the limitations section of the discussion:

“A further limitation is that many of the interventions consisted of provision of risk information alongside a range of additional information, either written or delivered in person or in groups. Separating the effect of the risk information from those additional elements of the interventions was therefore not possible. However, we chose not to exclude these studies from this review because it is unlikely that risk information would be incorporated into routine practice in isolation and, if anything, including them would overestimate the effect of the risk information.”

3. I am not sure why SMD (standardized mean difference) was used to pool perceived absolute risk or comparative risk. Risk is unit-free. Weighted mean difference should be good. Please describe. Also please be consistent with SMD. Sometimes standardized difference in means was used while in other places, standardized mean difference was used.

Response: As described above, the section of absolute and comparative risk has now been removed.

4. Abstract, please add description of perceived absolute risk and comparative risk and move the description of these terms in the methods. Also a summarized definition of all outcomes is necessary.

Response: As perceived absolute risk and comparative risk are no-longer outcomes of interest in this manuscript we have not added a description to the abstract.

5. Please clarify the difference for two sets of outcomes (perceived absolute/comparative risk vs. absolute/comparative risk (without perceived)). And please describe how studies measure absolute/comparative risk.

Response: This section has now been removed from the manuscript.

6. Why used “comparative risk”? “relative risk” is more common?

Response: These terms have been removed from the manuscript. In short though, we chose the term comparative risk as we believe it better reflects the risk being given which compares the individual to other individuals.

7. Introduction, please add “US” to the NCI. Not sure if other countries have NCI.

Response: We have added “US” as suggested.

8. Page 5 line 47, pre-post intervention studies are also observational studies. Please correct those in the exclusion (page 6 line 5).

Response: We thank the reviewer for bringing this error to our attention. We have, however, now restricted the review to randomised controlled trials. The sentence in the “Study selection” section now reads:

“Vignette, before-and-after studies without a control group, cross-sectional and qualitative studies were also excluded along with conference abstracts, editorials, commentaries and letters.”

9. Please add text to describe items in the quality assessment.

Response: As suggested, we have added the following text to describe the items in the quality assessment:

“We conducted quality assessment at the same time as data extraction using a checklist based on the Critical Appraisal Skills Programme (CASP) guidelines¹⁷ as an initial framework. This includes eight questions concerning whether the study addressed a clearly focused issue, the method of recruitment and randomisation, whether blinding was used, the measurement of the exposure and outcome, the comparability of the study groups and the follow-up.”

10. Page 10, line 40-44. Please describe the definition of “more accurate”. The authors had to dichotomize the outcomes to get pooled RR. Why not use RR in “perceived risk” page 8-9?

Response: We have removed this section from the manuscript.

Reviewer: 3

Reviewer Name: C Keeble

Institution and Country: University of Leeds, UK Competing Interests: None declared

This clearly explained review considers the impact of cancer risk information interventions in the adult population.

General comments:

-It is unfortunate that the pooling of results was limited, causing much of the article to be descriptive. Can the authors comment further on the data available for use in a meta-analysis?

Response: We agree that it is unfortunate that the pooling of results was limited. This was principally because there were either too few studies for each outcome or each study used different non-directly comparable measures. To make this clearer we have added the following text to the strengths and limitations section of the discussion:

“There was also large heterogeneity between the studies included and for all outcomes except attendance at screening there were either too few studies to meaningfully pool results or each study used different non-comparable measures.”

-Could the authors comment on the selection of the search terms used? There were over 30,000 articles returned, yet only 26 of these were included in the synthesis. An additional 6 were included following a handsearch outside of the initial 30,000.

Response: As suggested, we have added the following text to the Strengths and Limitations section of the discussion to reflect this.

“The main strengths of this review are the systematic search of multiple electronic databases and the broad inclusion criteria. Together these allowed us to include studies that assess the impact of interventions incorporating cancer risk information on multiple behavioural outcomes. However, from nearly 40,000 titles and abstracts we only included 14 with an additional 5 found through citation searching. This highlights the challenge in identifying studies in this area in which the primary purpose may not be related to the provision of risk information.”

-The review would benefit from further details of how the meta-analysis was conducted, with references for the method(s) and any Stata packages used.

Response: As suggested, we have added a reference for the method and details of the Stata package used as below:

“For this we used random effects meta-analysis¹⁸ and the ‘metan’ package in Stata.”

-A discussion of potential biases in the articles used would be beneficial, such as the types of people recruited. For example, were participants recruited at random from the population, or were they volunteers who could potentially be more likely to worry about their cancer risk?

Response: We agree with the reviewer that the types of people recruited is an important potential bias. We have accordingly provided additional details in Table 1 to clarify the sample from which participants were recruited and have added the following text to the results:

“Most recruited participants from those attending primary care clinics (n=3), or from lists of potentially eligible individuals from electronic medical records (n=7), telephone services (n=1), insurance records (n=1) or survey companies (n=1). Two recruited through schools, community centres and universities, one from those calling a cancer information service and three used public advertisements.”

We have also added the following text to the limitations section of the discussion:

“It is also possible that the findings do not reflect the potential impact of interventions incorporating information about cancer risk on general population as a whole: half of the included studies focused on female cancers and so only recruited women and all were subject to recruitment bias with the participants who agreed to take part potentially more interested in their cancer risk or more healthy, resulting in a bias in either direction.”

-Could the authors comment further on the moral obligation not to reduce worry, from a medical rather than psychological perspective.

Response: We agree with the reviewer that this is an interesting area, however having removed psychological outcomes from the manuscript we do not feel it is now relevant to the paper.

Minor comments:

- Hyphens may be useful in the title and introductory paragraphs (eg, risk-based).

Response: On the suggestion of the editorial team we have now replaced the term “risk-based” with “interventions incorporating information about cancer risk”.

-Many of the terms used (such as 'worry') may be subjective, can the authors comment on the potential weaknesses of subjective measures?

Response: We have now removed the section about worry from the manuscript.

-What role may the length of follow-up time in the studies have played? Could some studies be affected by short-term behavioral changes?

Response: We thank the reviewer for highlighting this important consideration. Of the 10 studies reporting health-related behaviours, five had a follow-up period of a year or more and three a period of six months. We therefore do not believe the studies were too short to detect changes in behaviour or reflected only immediate un-sustained changes. We have added the text below to the limitations section of the discussion to reflect this:

“The duration of follow-up also varied from 1 to 18 months. Although this makes pooling the findings more difficult, the studies with shorter follow-up were those with intention as the outcome measures and, of the 10 studies reporting health-related behaviours, five had a follow-up period of a year or more and three a period of six months. It is therefore unlikely that the studies as a whole were too short to detect changes in behaviour or reflected only immediate un-sustained changes.”

-Several of the cancers in the 32 articles were predominantly female cancers. Could gender have played a role in the findings of this review?

Response: We agree that as half of the included studies focused on female cancer, the findings may not be generalizable. We have added the following text to the discussion to reflect this:

“It is also possible that the findings do not reflect the potential impact of interventions incorporating information about cancer risk on general population as a whole: half of the included studies focused on female cancers and so only recruited women and all were subject to recruitment bias with the participants who agreed to take part potentially more interested in their cancer risk or more healthy, resulting in a bias in either direction.”

Reviewer: 4

Reviewer Name: Eva Lorenz

Institution and Country: Institut für Medizinische Biometrie, Epidemiologie und Informatik 55101 Mainz, Germany
Competing Interests: None declared

General comments

I was pleased to review this systematic review on the impact of cancer risk based interventions. I believe it is an important topic and will be of interest to many readers.

Response: We are pleased that the reviewer believes this is an important topic and will be of interest to many readers.

Comment: This systematic review and meta analysis has many strengths including a thorough search of literature, the different types of cancer and thereby interventions, and a variety of outcomes. One major methodology concern is the combination of diverse risk formats of the same studies in the pooled analyses. Since studies seem to be rather heterogeneous in the reporting of risk measures, the authors might remove or redesign the meta-analyses on 'risk perception and understanding of risk estimate' and on 'psychological well-being'. I would, in general, recommend removing the meta-analyses as the study heterogeneity between studies was rather high and the number of distinct studies in the separate analyses was rather low, e.g. only one study was considered for the pooled estimate of perceived comparative risk which is methodologically not meaningful.

Response: We agree with the reviewer. Finding ways to pool the findings on risk perception and understanding of the risk estimate and psychological well-being was a challenge. In light of the other comments made by the editorial team and the other reviewers, we have chosen to focus this revised manuscript instead on intention and behaviour. The meta-analyses referred to here have therefore been removed.

Comment: Title

Please elaborate on the term impact. What is actually the outcome of interest in this study?

Response: To clarify the outcome of interest we have amended the title to now read:

"Change in intention and behaviour following interventions incorporating information about cancer risk amongst the general population: a systematic review and meta-analysis of randomised controlled trials."

Comment: Abstract

I 13: Please specify the reason why the search started in 2000 and not earlier. This can either be done here or on page 5 in the section Search strategy.

Response: As suggested, we have added the reason our search began in 2000 in the Search strategy section of the methods as below:

"We chose to begin the search in 2000 as the previous review of tailored information about cancer risk and screening had noted that computer delivered interventions, as would be required for calculating risk scores, were only described in publications from 2000 onwards¹³."

There is no information on study heterogeneity given. Can you please add a sentence on the heterogeneity of studies (e.g. design, size) and outcomes?

I. 31: Please introduce the abbreviation 'RR' before first use.

Response: Done.

I. 36: Please harmonize the spelling of 'risk based' 'risk-based' throughout the manuscript (e.g. spelling in title differs from abstract).

Response: On the suggestion of the editorial team we have now replaced the term "risk-based" with "interventions incorporating information about cancer risk".

Box with strengths and limitations:

Please add: 'This systematic review and meta-analysis is the first...'

Response: Done.

Comment: The study heterogeneity is listed in this box but should already be mentioned in the abstract.

Response: As suggested we have added the sentence "There was significant heterogeneity in interventions and outcomes between studies." to the abstract.

Comment: How did the heterogeneity affect the pooling of results? Please specify.

Response: The heterogeneity meant that it was only possible to pool the results across the studies for the attendance at appropriate screening. For all the other outcomes there were either too few studies or each study used different non-directly comparable measures. To make this clearer in the strengths and limitations box we have amended the final bullet point to read:

"However, there was large heterogeneity across the studies and the different outcome measures included. This meant it was only possible to meta-analyse one outcome, attendance at screening."

Comment: Introduction

p. 5, I. 5: Please introduce RCT as an abbreviation for randomized controlled trial.

Response: Done.

Comment: The research question is not specifically defined. What do you mean by 'impact of the provision of cancer risk-based interventions'. Impact on what? Which different types of outcomes do you intend to investigate?

Response: We agree that in our review version of this manuscript the research question was too broad. We have therefore narrowed the focus of the review and changed the title and last paragraph of the discussion to reflect this as below:

Title – "Change in intention and behaviour following interventions incorporating information about cancer risk amongst the general population: a systematic review and meta-analysis of randomised controlled trials"

“Understanding the impact of interventions incorporating information about cancer risk on behaviour and intention to change behaviour before they are introduced into routine practice is important. Previous systematic reviews in this area have focused on trials in primary care¹², tailored information about cancer risk and screening^{13,14}. In this review we aimed to provide a comprehensive synthesis of the impact of interventions incorporating information about cancer risk on intention and behaviour within the general adult population.”

Methods

p.6, l.16: How was SGs rating incorporated in the rating? Did JUS and BS both review all articles or did they split the articles and then a random sample was reviewed by a third reviewer? Please specify if the first two reviewers did not individually read each of all articles.

Response: We are grateful to the reviewer for highlighting that we had not been completely clear here. JUS and BS each reviewed half of the papers and SG then reviewed 5% of those. The text has been amended to read:

“Two reviewers (JUS and BS) each screened half of the titles and abstracts to exclude papers that were clearly not relevant. A third reviewer (SG) independently assessed a random selection of 5% of the papers screened by each of the first reviewers.”

SG did not identify any papers for inclusion that had not already been identified by JUS or BS. The following text was already in the results section to describe that:

“After title and abstract screening by the first reviewers (JUS and BS), no additional papers met the inclusion criteria in the random 5% screened by the second reviewer (SG).”

p.6, l.56: Please introduce the abbreviation for Critical Appraisal Skills Programme before first use.

Response: Done.

p.7, l.38: Please remove: ‘we were only able to do this for a small number of outcomes’ to the results section.

Response: Done.

p. 7, l. 47: Please write Stata instead of STATA.

Response: Done.

p.8, l. 2: Same issue as previously: Did the first two reviewers share the screening or did they both read each article?

Response: As described above, we have amended the text in the methods section to clarify this.

Please add numbers of articles to the exclusion reasons.

Response: Done.

p.8, l.51: Consider adding percentages in order to know the relative proportion of 18 studies out of 32.

Response: We have now removed this section from the manuscript.

p.9, first paragraph: The 'format of the risk' is not introduced or described in detail. It is not clear to me as to why the authors report the estimates in e.g. the study by McCaul in 6 different groups (format of risk and time from intervention). Why are CR groups even included in the calculation of the perceived absolute risk? Each study should contribute once to the overall estimate for each format of risk category separately. Times from intervention are also not comparable and treating them equally should be avoided. If the difference post intervention and at baseline are to be pooled in the forest plot, each study should only occur once with one effect/difference estimate.

p.11, paragraph 'cancer worry': It seems as if you are again combining various effect estimates from overlapping study populations in one meta-analysis. Absolute and comparative risks cannot be treated as if they were the same and should be investigated in separate analyses. The intervention effects of the joint group AR+CR is always between the single AR and CR intervention effects. I assume this is true because the joint group includes the exact same individuals from both groups. If this is true, effect estimates cannot be seen as independent sources of information and cannot be combined in the analysis. The study by Helmes is reported twice with the exact same format of risk and time for intervention but with differing intervention effect. This seems to be a mistake.

Please replace the meta-analyses by analyses in subgroups where you specifically only compare AR or CR with each other but do not mix all different types in one pooled analysis since they are not comparable.

Response: As described above, for the reasons the reviewer mentions and in order to make the objective of the review clearer, these analyses have now been removed from the manuscript.

p.15, paragraph 'attendance at screening': You again excluded an article from the meta-analysis because no significant effect was observed. No presence of an effect is, however, a relevant piece of information and must be included in the analysis.

Response: The reason the article by Giles was excluded from the meta-analysis was because it did not include a control group. It was not because no significant effect was observed – 11 of the 12 studies in the meta-analysis had a significant effect either. As it was a cohort study it does not meet the revised inclusion criteria in this version and so reference to it has been removed.

The reason for excluding six studies from the analysis because of the absence of significant intervention effects is methodologically not tolerable. The strength of the intervention effects does not represent the actual intervention effect but a likely overestimation of effects. Non-significant effect estimates need to be considered in the analyses as well.

We are not clear which six studies the reviewer is referring to. We did not exclude any articles from the analysis because of the absence of significant intervention effects. As described above, in some cases it was not possible because of a lack of data to include all studies in the meta-analyses but these studies are then reported separately.

Figure 1: The PRISMA flow diagram should also report the number of articles considered for meta-analyses (by type of meta-analysis) and not only those considered in the review.

Response: As suggested, we have added a box reporting the number of articles included in the meta-analysis.

VERSION 2 – REVIEW

REVIEWER	Erika Waters WaMy research includes developing risk assessment and communication tools. shington University in St. Louis, USA
REVIEW RETURNED	17-Oct-2017

GENERAL COMMENTS	The authors have done a great job addressing the review, but I have a few additional comments. 1. The title is incorrect: Unless all of the articles used pre-post measures of intentions and behavior, the word "change" cannot be used. In addition, "information about cancer risk" is too vague and includes non-personalized intervention. An alternative would be "Effect of personalized cancer risk information on intentions and behavior: A systematic review..."2. The new sentence added to the end of the results in the abstract beginning "There is limited evidence that they increase intention to tan..." has 2 problems: First, i think they mean decrease intention to tan. Second, the entire sentence is confusing -- it might be missing some commas.3. p. 4--strengths and limitations box: the first bullet is incorrect. See DP French, Annals of Behavioral Medicine, October 2017 for a systematic review of systematic reviews of personalized risk communication interventions. Also, the 2nd bullet point needs to clarify that the cancer risk information is personalized. Otherwise it implies that studies that provided people with non-personalized lists of cancer risk factors are also being meta-analyzed. The problem of conflating "cancer risk information" with "personalized cancer risk information" occurs throughout the manuscript.4. JG Godino et al, PLOS Med, Nov 2016 should be included in the meta-analysis.5. I agree with the decision to remove the psychosocial variables (e.g., worry, decision conflict, QOL) and the information about changes/differences in absolute and comparative risk perceptions. However, I think the short, 1/2 paragraph on accuracy of risk perceptions should be retained (although NOT the paragraph about the impact of formats on accuracy). Accuracy is important because the authors' assertion, in the discussion and elsewhere, that personalized risk assessment tools can help people make informed decisions rests implicitly on the idea that people need an accurate idea of what their risk is. This is demonstrated explicitly on p. 25 beginning "This is important as on average many of the general population overestimate their own risk of cancer...." I strongly suggest adding back in the 6 RCTs that reported accuracy of risk perceptions with and without risk information.6. The 1st sentence of the discussion has 2 points of confusion: (1) what do the authors mean by "in all settings", and (2) the sentence beginning "there is limited evidence" has the same error as I identified in the abstract -- "that they increase intention to tan..."
--

REVIEWER	Zhen Wang Mayo Clinic, USA
REVIEW RETURNED	13-Oct-2017

GENERAL COMMENTS	The authors have addressed all of concerns. I have no more comments.
--

REVIEWER	C Keeble University of Leeds, UK
REVIEW RETURNED	31-Oct-2017

GENERAL COMMENTS	The authors have satisfactorily addressed each of my previous points. I have no further comments.
---

REVIEWER	Eva Lorenz Institute for Medical Biometry, Epidemiology and Informatics (IMBEI), Mainz, Germany
REVIEW RETURNED	26-Oct-2017

GENERAL COMMENTS	General comments I was pleased to see that the authors addressed the previous comments in sufficient detail. The manuscript improved a lot since the last revision and I only have minor suggestions for revision. Keywords Cancer risk should be one keyword and not separated by a comma. If the meta-analysis remains in the article then it should be listed in the keywords. Strengths and limitations of the study First bullet point: Intention and behavior of individuals regarding what? This should be specified. Data synthesis and statistical analysis We present intervention effects as relative risk rather than odds ratios to avoid overestimating the risk²⁰. ⇒ Please use the abbreviations for RR. Results Unfortunately, I was not able to have a look at Figure 2 in order to assess the results from the meta-analysis in detail, since it is not part of the Supplemental files provided for this review. Discussion Please elaborate a bit more on the limitations of the meta-analysis. Are differences in the attendance at screening expected by cancer type? Were the estimates adjusted for further covariates or is the analysis based on crude estimates?
--

VERSION 2 – AUTHOR RESPONSE

Reviewer: 1

Reviewer Name: Erika Waters

Institution and Country: Washington University in St. Louis, USA Competing Interests: My research includes developing risk assessment and communication tools.

The authors have done a great job addressing the review, but I have a few additional comments.

We are pleased you feel we have addressed your previous comments. Thank you again for those. We have addressed your additional comments below.

1. The title is incorrect: Unless all of the articles used pre-post measures of intentions and behavior, the word "change" cannot be used. In addition, "information about cancer risk" is too vague and includes non-personalized intervention. An alternative would be "Effect of personalized cancer risk information on intentions and behavior: A systematic review..."

Response: We agree with the reviewer that her suggested alternative title is clearer. As the editorial team mentioned in the previous revision we believe it is important to retain the phrase "interventions incorporating" to reflect the fact that many of the interventions also have other components. We have therefore amended the title to be "Effect of interventions incorporating personalised cancer risk information on intentions and behaviour: a systematic review and meta-analysis of randomised controlled trials"

2. The new sentence added to the end of the results in the abstract beginning "There is limited evidence that they increase intention to tan..." has 2 problems: First, i think they mean decrease intention to tan. Second, the entire sentence is confusing -- it might be missing some commas.

Response: Thank you for noticing our error in the results section. As you suggest, it should have read "decrease intention to tan". We have split the sentence into two so that it now reads:

"There is limited evidence that they increase smoking abstinence, sun protection, adult skin self-examination and breast examination and decrease intention to tan. However, they do not increase smoking cessation, parental child skin examination or intention to protect skin."

3. p. 4--strengths and limitations box: the first bullet is incorrect. See DP French, Annals of Behavioral Medicine, October 2017 for a systematic review of systematic reviews of personalized risk communication interventions. Also, the 2nd bullet point needs to clarify that the cancer risk information is personalized. Otherwise it implies that studies that provided people with non-personalized lists of cancer risk factors are also being meta-analyzed. The problem of conflating "cancer risk information" with "personalized cancer risk information" occurs throughout the manuscript.

Response: We believe this is the first systematic review to synthesize evidence on the effect of interventions incorporating personalised cancer risk information on intentions and behaviour across multiple settings. The review of reviews by French et al that you mention includes nine reviews. Of those only one focuses on cancer and that reviewed the effectiveness of testing for genetic susceptibility to smoking-related diseases on smoking cessation outcomes (Smerecnik et al 2012). As our review focuses on provision of personal estimates of future cancer risk based on two or more non-genetic variables, the studies included here do not overlap with those included in French et al's review. As we mention in the introduction, three other systematic reviews have included studies providing cancer risk information but they have focused only on trials in primary care or tailored

information about cancer risk and screening. In response to your point 6 we have, however, clarified that this review includes studies in which the risk information is delivered across all settings. The first bullet point now reads:

“This systematic review is the first comprehensive review of the effect on intention and health-related behaviour of individuals in the general population of interventions delivered across multiple settings which incorporate personalised information about cancer risk.”

As suggested, we have added the word “personalised” in the second bullet point so that it reads:

“The use of a broad search strategy across multiple databases enabled us to identify 19 randomised controlled trials reporting the impact of interventions incorporating personalised cancer risk information on 12 outcomes.”

We have been through the whole manuscript and added the word “personalised” before cancer risk to make this clearer.

4. JG Godino et al, PLOS Med, Nov 2016 should be included in the meta-analysis.

Response: The paper you refer to by Godino et al was a trial of genetic or phenotypic risk estimates for type 2 diabetes. As this review is limited to those studies in which estimates of cancer risk were provided it does not meet the inclusion criteria. We have therefore not included it.

5. I agree with the decision to remove the psychosocial variables (e.g., worry, decision conflict, QOL) and the information about changes/differences in absolute and comparative risk perceptions. However, I think the short, 1/2 paragraph on accuracy of risk perceptions should be retained (although NOT the paragraph about the impact of formats on accuracy). Accuracy is important because the authors' assertion, in the discussion and elsewhere, that personalized risk assessment tools can help people make informed decisions rests implicitly on the idea that people need an accurate idea of what their risk is. This is demonstrated explicitly on p. 25 beginning "This is important as on average many of the general population overestimate their own risk of cancer...." I strongly suggest adding back in the 6 RCTs that reported accuracy of risk perceptions with and without risk information.

Response: We are pleased that you agree with our decision to remove the psychosocial variables from this review. We also agree with you that understanding the impact of personalised risk estimates on accuracy is important. However, as this review now focuses on the impact of personalised risk estimates on intention and health-related behaviours we do not feel it is appropriate to include the 6 RCTs that report accuracy of risk perception.

Risk accuracy is also only one measure of risk perception and we believe including that without the information about changes/differences in absolute and comparative risk perceptions would be oversimplifying the association. We do intend to report effects on psychosocial variables and risk perceptions, including risk accuracy, in a separate paper.

6. The 1st sentence of the discussion has 2 points of confusion: (1) what do the authors mean by "in all settings", and (2) the sentence beginning "there is limited evidence" has the same error as I identified in the abstract -- "that they increase intention to tan..."

Response: We had included the phrase “in all settings” to reflect the inclusion of studies in which the personalised cancer risk information had been given in any setting, not just limited to primary or secondary care. We agree this was confusing as originally written so have amended the sentence. We have also amended the third sentence as above. The section now reads:

“This systematic review is, to our knowledge, the first review of the impact of interventions delivered across multiple settings which incorporate personalised information about cancer risk on intention and behaviour in the general population. The findings show that such interventions do not affect intention to attend or attendance at screening. There is limited evidence that they increase smoking abstinence, sun protection, adult skin self-examination and breast examination and decrease intention to tan. However, this was not seen for smoking cessation, parental child skin examination or intention to protect skin.”

Reviewer: 2

Reviewer Name: Zhen Wang

Institution and Country: Mayo Clinic, USA Competing Interests: None declared

The authors have addressed all of concerns. I have no more comments.

Response: We are pleased that you feel we have addressed all your previous concerns. Thank you again for your earlier comments.

Reviewer: 3

Reviewer Name: C Keeble

Institution and Country: University of Leeds, UK Competing Interests: None declared

The authors have satisfactorily addressed each of my previous points. I have no further comments.

Response: We are pleased that you feel we have addressed all your previous concerns. Thank you again for your earlier comments.

Reviewer: 4

Reviewer Name: Eva Lorenz

Institution and Country: Institute for Medical Biometry, Epidemiology and Informatics (IMBEI), Mainz, Germany Competing Interests: None declared

General comments

I was pleased to see that the authors addressed the previous comments in sufficient detail. The manuscript improved a lot since the last revision and I only have minor suggestions for revision.

Response: We are pleased that you feel the manuscript has improved. We have addressed each of your additional suggestions below.

Comment: Keywords

Cancer risk should be one keyword and not separated by a comma. If the meta-analysis remains in the article then it should be listed in the keywords.

Response: As suggested we have removed the comma between cancer and risk in the list of keywords and have added meta-analysis.

Comment: Strengths and limitations of the study

First bullet point: Intention and behavior of individuals regarding what? This should be specified.

Response: As suggested we have clarified that the behaviour of interest is health-related behaviour. The bullet point now reads:

“This systematic review is the first comprehensive review of interventions incorporating cancer risk on intention to change health-related behaviours and health-related behaviours themselves in individuals in the general population.”

Comment: Data synthesis and statistical analysis

We present intervention effects as relative risk rather than odds ratios to avoid overestimating the risk²⁰.

⇒ Please use the abbreviations for RR.

Response: We have added the abbreviations for RR

“We present intervention effects as relative risk (RR) rather than odds ratios (OR) to avoid overestimating the risk”

Comment: Results

Unfortunately, I was not able to have a look at Figure 2 in order to assess the results from the meta-analysis in detail, since it is not part of the Supplemental files provided for this review.

Response: We are sorry that you were unable to see Figure 2. We have included it below for your reference.

Comment: Discussion

Please elaborate a bit more on the limitations of the meta-analysis. Are differences in the attendance at screening expected by cancer type? Were the estimates adjusted for further covariates or is the analysis based on crude estimates?

Response: As you suggest, the estimates in the meta-analysis for screening attendance is based on crude estimates. To address this and other limitations we have added the following text to the discussion:

“There was also significant heterogeneity in the outcome measures included, duration of follow-up and method of recruitment across the included studies. For all outcomes except attendance at screening there were either too few studies to meaningfully pool results or each study used different non-comparable measures. Even for attendance at screening for which meta-analysis was possible, we were only able to pool crude estimates and the included studies addressed screening for breast, bowel and cervical cancer. While it is possible that the impact on screening attendance might be different across the different cancer sites because of the nature of the tests involved, the finding that only one study of mammography showed an effect of interventions incorporating personalised cancer risk information suggests that this is unlikely to be the case.”

VERSION 3 – REVIEW

REVIEWER	Eva Lorenz Institute for Medical Biometry, Epidemiology and Informatics (IMBEI), Mainz, Germany
REVIEW RETURNED	21-Nov-2017
GENERAL COMMENTS	The authors have done a great job addressing my last review and I have no further comments.